# Are Caffeine’s Effects on Resistance Exercise and Jumping Performance Moderated by Training Status?

**DOI:** 10.3390/nu14224840

**Published:** 2022-11-16

**Authors:** Erfan Berjisian, Alireza Naderi, Shima Mojtahedi, Jozo Grgic, Mohammad Hossein Ghahramani, Raci Karayigit, Jennifer L. Forbes, Francisco J. Amaro-Gahete, Scott C. Forbes

**Affiliations:** 1Department of Exercise Physiology, Faculty of Physical Education and Sport Sciences, Tehran University, Tehran 1417935840, Iran; 2Department of Exercise Physiology, Borujerd Branch, Islamic Azad University, Borujerd 6915136111, Iran; 3Institute for Health and Sport, Victoria University, Melbourne, VIC 3011, Australia; 4Physical Education and Sport Sciences Department, Faculty of Humanities, Tarbiat Modares University, Tehran 1411713116, Iran; 5Faculty of Sport Sciences, Ankara University, Gölbaşı, Ankara 06830, Turkey; 6Department of Physical Education Studies, Brandon University, Brandon, MB R7A 6A9, Canada; 7PROmoting FITness and Health through Physical Activity Research Group (PROFITH), Department of Physical and Sports Education, Faculty of Sports Science, University of Granada, 18011 Granada, Spain; 8Department of Physiology, Faculty of Medicine, EFFECTS-262 Research Group, University of Granada, 18016 Granada, Spain; 9Centro de Investigación Biomédica en Red Fisiopatología de la Obesidad y Nutrición (CIBERobn), Instituto de Salud Carlos III, 28029 Madrid, Spain

**Keywords:** caffeine, ergogenic aid, resistance training, muscular strength-endurance

## Abstract

This study aimed to explore if the effects of caffeine intake on resistance exercise and jumping performance are moderated by training status. We included ten resistance-trained and ten recreationally active males in a randomized, double-blind, crossover study. Participants were categorized into groups according to their resistance to training experience and muscular strength levels. Exercise performance outcomes included weight lifted and mean velocity during a one-repetition maximum (1RM) bench press and squat; repetitions were performed to muscular failure in the same exercises with 70% of 1RM and countermovement jump (CMJ) height. Exercise performance was evaluated on three occasions, following no substance ingestion (control), caffeine (6 mg/kg), and placebo. There was a main effect on the condition for all the performance outcomes (all *p* ≤ 0.02), except for the 1RM squat mean velocity (*p* = 0.157) and 1RM bench press mean velocity (*p* = 0.719). For weight lifted in the 1RM bench press, there was a significant difference when comparing the caffeine vs. control, caffeine vs. placebo, and placebo vs. control. For weight lifted in the 1RM squat, a significant difference was found when comparing the caffeine vs. control. For muscular endurance outcomes and jump height, a significant difference was found when caffeine was compared to the control or placebo. Effect sizes were trivial for muscular strength (Hedges’ *g*: 0.04–0.12), small for the jump height (Hedges’ *g*: 0.43–0.46), and large for muscular endurance (Hedges’ *g*: 0.89–1.41). Despite these ergogenic effects, there was no significant training status × caffeine interaction in any of the analyzed outcomes. In summary, caffeine ingestion is ergogenic for muscular strength, endurance, and jump height. These effects are likely to be of a similar magnitude in resistance-trained and recreationally active men.

## 1. Introduction

Caffeine is one of the most popular and investigated ergogenic aids [1]. Caffeine supplementation has been reported as ergogenic for outcomes such as aerobic endurance, power, and performance in various sport-specific tasks [1,2]. In resistance exercise, the available studies indicate that caffeine ingestion increases muscular strength and endurance, as well as peak and mean repetition velocity [3,4]. These ergogenic effects are likely explained by caffeine’s ability to bind to adenosine A_1_ and A_2_ receptors. After binding to these receptors, caffeine may promote wakefulness and alleviate fatigue sensations, which may contribute to improvements in exercise performance [5]. Additionally, caffeine may enhance motor unit recruitment and reduce the decline in voluntary activation occurring due to exercise-induced fatigue, effects which may also contribute to performance improvements [6,7]. 

While an ergogenic effect in caffeine is commonly observed, studies that plot individual participant data show considerable between-individual variation in response to caffeine ingestion [8,9]. For example, one study presented individual participant data regarding the effects of caffeine intake on one-repetition maximum (1RM) strength in a sample of 17 participants [8]. Out of this sample, twelve recorded higher strength values following caffeine ingestion, three participants performed better in the placebo trial, and the performance of two participants was unaffected by supplementation. Several aspects have been identified as potential modulators of the individual response to the effects of caffeine, including habitual caffeine intake, variations in genotypes, and training status [10]. While many studies focused on habitual caffeine intake and genotype variations, the evidence on training status-specific responses is scarce [11,12]. Training status is proposed as a potential moderator of caffeine’s effects due to methodological and physiological factors. From a methodological standpoint, some researchers have hypothesized that trained individuals exhibit greater reliability in different exercise tasks, thus decreasing the risk of type II errors and preventing possible false-negative findings [13]. From a physiological standpoint, it is shown that trained individuals have greater adenosine A_2a_ receptor densities [14]. This may be relevant as caffeine’s effects are mostly explained by the adenosine receptor antagonism. Thus, caffeine may be more ergogenic in trained individuals due to its greater binding to these receptors mediated by their greater densities. 

The first study that directly compared the effects of caffeine in populations with different training statuses reported an ergogenic effect on swimming velocity in highly trained swimmers but not in recreationally trained ones [15]. Astorino et al. also showed that caffeine ingestion improved performance in a 10 km cycling time trial in endurance-trained but not “active” participants [16]. However, subsequent research did not replicate these findings, as other studies reported similar responses to caffeine ingestion in trained and untrained individuals [17,18,19]. Most studies on this topic used different endurance tasks to evaluate performance, while the effects of training status on responses to caffeine ingestion in resistance exercise remain largely unexplored. Thus far, only one such study has been performed [20]. In contrast to previous findings, this study reported that caffeine and placebo ingestion enhanced 1RM squat strength only in untrained individuals [20]. In summary, while training status might influence the ergogenic effects of caffeine, the currently available evidence is either conflicting (i.e., in opposite directions) or limited—at least for outcomes related to resistance exercise.

Given the outlined limitations in the current body of evidence, this study aimed to explore caffeine’s effects on muscular strength, endurance, and power in resistance-trained vs. recreationally active men. We hypothesized that caffeine ingestion would be ergogenic for all outcomes in both trained and recreationally active men.

## 2. Materials and Methods

### 2.1. Participants

A total of 20 healthy, trained (*n* = 10), and recreationally active (*n* = 10) men participated in this study (trained: age: 31 ± 6 years; body mass: 78.7 ± 10.4 kg; height: 176 ± 7 cm; body mass index: 25.3 ± 2.9; recreationally active: age: 24 ± 9 years; body mass: 71.6 ± 12.7 kg; height: 174 ± 6 cm; body mass index: 23.6 ± 4.0). Based on the recent classification of training status, we categorized ten participants as “trained,” given that they satisfied the criteria of regularly training three or more times per week with the purpose of competing in a specific sport (i.e., bodybuilding, volleyball, soccer) [21]. Our trained participants also had a minimum of five years of resistance training experience and were regularly performing this type of exercise (≥4 h per week) during the study. To be included in the trained group, participants also had to satisfy specific muscular strength criteria, i.e., possess the ability to (at least) perform the 1RM squat using a load 1.5 times greater than their body weight and perform a 1RM bench press with a load equal to their body weight [22]. The recreationally active participants had less than six months of resistance training experience and performed three or fewer hours of this type of activity per week. The participants did not report any injury in the last six months before the study and did not use any ergogenic aids in the last three months before the data collection. Participants were low habitual caffeine consumers (<50 mg/day), as determined by a validated questionnaire [23]. Participants were informed of the research’s potential risks, benefits, and dissemination of its findings before providing written informed consent to participate. The study’s procedures met the latest revised Declaration of Helsinki [24]. The Tehran University Ethics Committee provided ethical approval for this study.

### 2.2. Experimental Design

In a randomized, double-blind, crossover study design, participants completed three testing conditions: caffeine, placebo, and control. The participants were randomly assigned to the three conditions using the Latin Square model [25] and the Research Randomizer software (www.randomizer.org; accessed on 10 May 2022). An independent researcher conducted all randomization procedures to ensure a double-blind study design. Participants attended the laboratory four times, including one familiarization session and three experimental sessions, with at least 72 h of washout period between sessions. To eliminate any circadian effects, all assessments were conducted at the same time of the day (between 12:00 and 16:00 h). Additionally, the participants were requested to keep their dietary intake in the 24 h before each testing session consistent and to refrain from strenuous exercise within the 48 h before testing. On testing days, the participants were required to refrain from caffeine intake. A standardized pre-workout snack consisting of 1.5 g/kg of carbohydrates and 20 g of protein was provided three to four hours before starting the test [26]. Each participant ingested one capsule containing 6 mg/kg body mass of pure caffeine anhydrous (Cat. No. C0750; Sigma-Aldrich; Steinheim; Germany) or a placebo (cellulose) 60 min before starting the warm-up [27]. A high-precision electronic digital scale was used to weigh the caffeine and placebo powder, which was administered to capsules by an independent researcher. In the control trial, the participants also came to the gym 60 min before exercise but did not ingest any capsules. 

### 2.3. Exercise Protocol

#### 2.3.1. Warm-Up

The participants first performed a warm-up, which included light running on a treadmill and self-selected general upper or lower-body warm-up exercises. As the warm-up was self-selected, the participants were required to replicate the warm-up used in the first session in all the remaining sessions. 

#### 2.3.2. 1RM Testing

Following the warm-up, 1RM testing was performed. The 1RM testing was initially performed for the bench press, followed by the squat. Of note, following the 1RM bench press, the participants first rested and then performed a muscular endurance assessment in the same exercise before starting with the 1RM squat (Figure 1). The 1RM testing was performed according to the National Strength and Conditioning Association Guidelines [28]. For the 1RM test, the first warm-up set consisted of 6–8 repetitions, using 50% of the 1RM determined during the familiarization session. The second and third sets involved 4 and 3 repetitions with 70% and 80% of the 1RM, respectively. In the fourth set, the participants performed one repetition with 95% of 1RM. Then, 1RM attempts were performed. In the case of a successful 1RM, the weight was further increased by 2.5 kg until the participant could no longer record a successful attempt. If the 1RM attempt was unsuccessful, the load was decreased by 2.5 kg for further attempts until a successful one was recorded. Five-minute rest intervals were allowed between each 1RM attempt. 

#### 2.3.3. 1RM Mean Repetition Velocity

In addition to evaluating the maximum lifted load in each 1RM, we also assessed the mean repetition velocity using a mobile phone application (*PowerLift*) [29]. This application has a high test-retest reliability, as demonstrated by the intra-class correlation coefficient that ranged from 0.93 to 0.99 for the mean repetition velocity [29]. Using this application, we were able to: (a) record the video of the exercise in slow motion, (b) analyze the video material frame-by-frame, and (c) manually select the beginning and the end of the concentric phase of the repetition. The application calculated the time (in m/s) between two phases and provided mean repetition velocity data [29]. For the bench press, the push-off phase was considered the beginning of the concentric movement. The end of the movement was considered the moment when the participants fully extended their elbows. For the squat, the beginning of the concentric phase was considered when the participant started extending in the knee joint from the bottom phase. The end of the movement was considered when the participant fully extended at the knee and hip joint and was in an upright position. 

#### 2.3.4. Muscular Endurance Assessment 

After 1RM determination for each exercise, five minutes of rest were provided to the participants. Following the rest interval, they performed a single set to muscular failure using 70% of 1RM. In both the bench press and squat, the participants were required to keep the repetition tempo consistent without any between-repetition rest. The outcome of the muscular endurance assessment was the number of repetitions performed. A strength and conditioning coach monitored the techniques for each exercise. 

#### 2.3.5. Countermovement Jump

After the muscular endurance assessment in the squat exercise, participants were given two minutes of rest. They then performed a countermovement jump (CMJ) test, which included three maximum attempts with a one-minute rest between each jump. Each jump was recorded using a validated mobile phone application (*My jump)*. With this application, we were able to record the entire movement and select the take-off (no feet on the ground) and landing (at least one foot on the ground) phases of the jump. Based on these inputs, the application provides the jump height performance. The highest jump was used for subsequent data analysis. For the CMJ, an experienced researcher instructed participants on the correct CMJ technique using video and live demonstrations during the familiarization session. Participants were asked to stand with a straight torso and knees fully extended with the feet shoulder-width apart and then execute a quick downward movement (to approximately 90° at the knees), followed by a fast-upward movement without an arm swing. During the familiarization session, the participants practiced the CMJ while a researcher corrected their technique where necessary. 

### 2.4. Rating of Perceived Exertion

The rating of perceived exertion (RPE) was evaluated at the end of each session, using the 0 to 10 scale. We only evaluated the session RPE. In other words, the participants were required to rate their overall perceived exertion during the whole session, not just for a specific exercise. 

### 2.5. Effectiveness of the Blinding

To evaluate the effectiveness of the blinding, we asked the participants to respond to the following question: “Which supplement do you think you have ingested?” [30]. The question had three possible responses: (a) “caffeine”, (b) “placebo”, and (c) “I do not know” [30]. If the participants responded with “a” or “b”, they were required to express the reason for selecting their response. This assessment was performed before and after exercise only in the caffeine and placebo trials.

### 2.6. Statistical Analysis

An independent *t*-test was used to compare the age, height, body mass, and body mass index between the trained and recreational groups. We used a two-way repeated measure ANOVA to the test training status (resistance-trained vs. recreationally active) × caffeine (caffeine vs. placebo vs. control) interaction on the performance and RPE data. In the case of a significant main effect, Tukey’s post hoc pairwise comparison test was used to determine where the difference occurred. Effect sizes were calculated using Hedge’s *g* for repeated measures. Hedges’ *g* values of <0.20, 0.20 to 0.49, 0.50 to 0.79, and ≥0.80 were considered to represent trivial, small, moderate, and large effects, respectively. The statistical significance was set at *p* < 0.05. Statistical analyses were performed using Statistica version 14.0.0.15.

## 3. Results

There were no significant differences between the trained and recreational groups for age, height, body mass, or body mass index (*p* > 0.05). 

### 3.1. Bench Press

For the weight lifted in the 1RM bench press, we did not find a significant training status × caffeine interaction (*p* = 0.823). There was a significant main effect of caffeine (*p* < 0.001). Pairwise comparisons revealed a significant difference between the caffeine vs. placebo (absolute ∆ = 2.9 ± 3.3 kg; *p* < 0.001), caffeine vs. control (absolute ∆ = 4.4 ± 1.8 kg; *p* < 0.001), and placebo vs. control (absolute ∆ = 1.5 ± 2.7 kg; *p* = 0.049) (Table 1 and Table 2). There was a significant effect in the training status (*p* < 0.001), with higher strength values observed in the trained group (Table 1).

For 1RM bench press mean velocity, we did not find any significant differences (interaction: *p* = 0.502; caffeine: *p* = 0.719; training status: *p* = 0.143).

For the number of repetitions performed with 70% of 1RM, we did not find a significant training status × caffeine interaction (*p* = 0.218). There was a significant main effect of caffeine (*p* < 0.001). Pairwise comparisons revealed a significant difference between the caffeine vs. placebo (absolute ∆ = 4.5 ± 2.1 repetitions; *p* < 0.001) and caffeine vs. control (absolute ∆ = 4.1 ± 2.3 repetitions; *p* < 0.001). We did not find a significant effect on the training status (*p* = 0.647).

### 3.2. Back Squat 

For the weight lifted in the 1RM squat, we did not find a significant training status × caffeine interaction (*p* = 0.572). There was a significant main effect of caffeine (*p* = 0.020). Pairwise comparisons revealed a significant difference only between the caffeine vs. control (absolute ∆ = 2.6 ± 3.4 kg; *p* = 0.015). There was a significant effect on the training status (*p* < 0.001), with higher strength values observed in the trained group (Table 1). 

For the 1RM back squat mean velocity, we did not find a significant training status × caffeine interaction (*p* = 0.811) or a significant main effect of caffeine (*p* = 0.157). There was a significant effect on the training status (*p* = 0.025), with higher velocity values observed in the recreationally active group.

For the number of repetitions performed with 70% of 1RM, we did not find a significant training status × caffeine interaction (*p* = 0.954). There was a significant main effect of caffeine (*p* < 0.001). Pairwise comparisons revealed a significant difference between the caffeine vs. placebo (absolute ∆ = 4.1 ± 4.7 repetitions; *p* < 0.001) and caffeine vs. control (absolute ∆ = 4.9 ± 3.6 repetitions; *p* < 0.001). We did not find a significant effect on the training status (*p* = 0.430). 

### 3.3. CMJ

For CMJ height, we did not find a significant training status × caffeine interaction (*p* = 0.933). There was a significant main effect of caffeine (*p* < 0.001). Pairwise comparisons revealed a significant difference between the caffeine vs. placebo (absolute ∆ *=* 3.8 *±* 3.1 cm; *p* < 0.001) and caffeine vs. control (absolute ∆ *=* 3.9 ± 3.4 cm; *p* < 0.001). There was a significant effect on the training status (*p* < 0.001), with higher jump values observed in the trained group (Table 1).

### 3.4. RPE

For session RPE, we did not find any significant differences (interaction: *p* = 0.374; training status: *p* = 0.303; condition: *p* = 0.054). 

### 3.5. Effectiveness of the Blinding

The placebo trials were correctly identified by 30% of the participants. Caffeine trials were correctly identified by 20%—40% of participants (Table 3).

## 4. Discussion

The main finding of the present study indicates that acute caffeine ingestion enhances muscular strength, endurance, and jump height. While an ergogenic effect of caffeine was observed, we did not find significant differences between the participants varying in training status. These results potentially suggest that training status may not moderate caffeine’s ergogenic effects on resistance exercise and jumping performance.

The effects of caffeine on athletes with different fitness levels have been previously investigated in a limited number of studies with contrasting results [16,17,18,19,20,31]. However, only one study explored the effects of caffeine with participants stratified according to training status on resistance exercise outcomes. Brooks et al. [20] examined the 1RM squat performance in a controlled trial or after consuming 5 mg/kg of caffeine or a placebo in 14 participants (seven trained and seven untrained). Both caffeine and placebo ingestion increased the 1RM squat strength when compared to the control, but only in the untrained group. Several differences between our study and the work by Brooks et al. [20] exist that might explain these divergent findings. First, these authors classified participants as trained if they were performing resistance training at least 3 days a week for the past 6 months. For the untrained group, it was stated that these participants did not perform any resistance training in the past 6 months. However, the strength values observed in the untrained group may question whether this group was indeed “untrained”. Specifically, their 1RM in the squat ranged from 1.2 to 1.4 times their body weight. This is notable to mention, given that classifications provided by other authors may consider the training status of these participants as intermediate-to-advanced [32]. In our study, the recreationally active participants had a 1RM in the squat similar to or lower than their body weight, demonstrating the differences in training status between the studies. Secondly, Brooks et al. [20] found ergogenic effects for the caffeine and placebo vs. the control, but no significant differences between caffeine and placebo. As there were no significant differences between the caffeine and placebo, these results suggest that performance improvements are due to the placebo effect, not necessarily the physiological effects of the caffeine.

It was hypothesized that caffeine might elicit a larger ergogenic effect in the trained participants due to their greater reliability of exercise performance or higher density of adenosine receptors [10]. Studies that evaluated the test-retest reliability of various exercise assessments commonly reported similar reliability values in trained and untrained individuals [33,34,35]. For example, a recent review focused on the test-retest reliability of the 1RM strength assessment and reported that 92% and 93% of the intra-class correlation coefficients were ≥0.90 for untrained and trained participants, respectively [36]. These data illustrate that the test-retest reliability for the 1RM test is nearly identical in both populations. Similar findings have been observed for various jumping outcomes [37]. Besides reliability, it was hypothesized that a greater effect of caffeine might occur in trained individuals due to their greater density of adenosine receptors [10]. However, the veracity of this hypothesis may also be questionable, given that the differences in adenosine receptor density in the trained and untrained populations were only observed for cardiac and triceps brachii muscles [14]. This is relevant to mention as caffeine mainly acts by binding to adenosine receptors in the brain, not necessarily in the skeletal muscle [5]. Finally, animal model studies have reported that the application of caffeine enhances muscle power similarly in mice that either underwent an 8-week exercise intervention or those that were selected to act as an untrained control [38]. Overall, it seems that caffeine’s effects on resistance exercise are not modulated by training status, even though much more research in this field is needed.

In most analyzed outcomes, there was a main effect of caffeine. These results further confirm the ergogenic effect of caffeine on muscular strength, endurance, and jump height. Currently available meta-analytical data have reported that caffeine ingestion enhances muscular strength and jump height by a trivial-to-small magnitude, while its effects on muscular endurance tend to be higher [39,40]. Indeed, we reported trivial effect sizes for muscular strength (Hedges’ *g*: 0.04–0.12) and a small trivial effect for the jump height (Hedges’ *g*: 0.43–0.46). For muscular endurance, Hedges’ *g* was from 0.89 to 1.41, which is in the range of effect sizes previously reported for this outcome [3]. Interestingly, we did not find a significant main effect of caffeine on RPE values, suggesting that caffeine’s ergogenic effects could be explained by other factors, such as increased motor unit recruitment and an attenuated decline in voluntary activation during exercise [7].

While caffeine was generally ergogenic compared to the placebo and control, we also found a significant difference between the placebo and control for the 1RM bench press. These results add to the growing body of evidence, which indicates that a certain proportion of the ergogenic effect of caffeine is attributed to placebo effects [41]. These results are in line with a recent meta-analysis reporting that around 50% of the ergogenic effect of caffeine is attributed to a placebo [41]. However, it should also be stated that caffeine’s effects in the present study were much more consistent than the placebo’s, given that they were observed in most analyzed outcomes.

When interpreting the findings herein, there are several limitations that need to be considered. We binarily categorized participants into two groups, trained and recreationally active. While such an approach was used in similar studies on the topic, it inherently offers limitations due to the strict criteria used for classification. Specifically, while we did not find that training status modulated the effects of caffeine, it might be possible that highly trained athletes may not benefit from caffeine ingestion, as their performance is already at a high level and might be approaching their genetic ceiling. However, in the present study, training status did not seem to alter the response to caffeine. Additionally, we included only young male participants; therefore, these results may not necessarily be generalized to females or older adults.

## 5. Conclusions

The main finding of the present study indicates that acute caffeine ingestion enhances muscular strength, endurance, and jump height. The magnitude of caffeine’s effects was trivial, small, and large for muscular strength, jump height, and muscular endurance, respectively. While an ergogenic effect of caffeine was observed, we did not find significant differences between the two groups of varying training statuses. These results potentially suggest that training status may not moderate caffeine’s ergogenic effects on resistance exercise and jumping performance.

## Figures and Tables

**Figure 1 nutrients-14-04840-f001:**
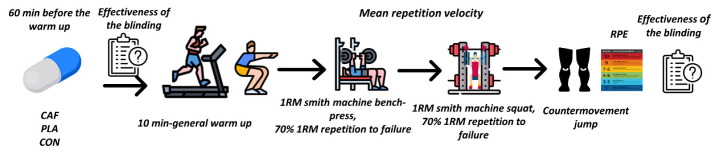
Schematic representation of the study design. CAF: caffeine. PLA—placebo. CON—control. 1RM—one-repetition maximum. RPE—rating of perceived exertion.

**Table 1 nutrients-14-04840-t001:** Summary of the performance and perceptual responses in the three trials.

Variable	Trained (Caffeine)	Trained (Placebo)	Trained (Control)	Recreationally Active (Caffeine)	Recreationally Active (Placebo)	Recreationally Active (Control)	Interaction (*p*-Value)	Condition (*p*-Value)	Training Status (*p*-Value)
1RM bench press (kg)	114.5 ± 18.3	111.5 ± 15.6	109.8 ± 17.0	52.8 ± 8.9	50.0 ± 8.3	48.8 ± 8.8	0.823	<0.001	<0.001
1RM bench press mean velocity (m/s)	0.15 ± 0.04	0.15 ± 0.03	0.16 ± 0.03	0.14 ± 0.02	0.15 ± 0.02	0.14 ± 0.01	0.502	0.719	0.143
Repetitions to failure with 70% of 1RM in the bench press (repetitions)	19.4 ± 3.4	15.1 ± 2.3	16.1 ± 2.6	20.6 ± 3.3	16.0 ± 3.1	15.7 ± 3.1	0.218	<0.001	0.647
1RM Squat(kg)	155.0 ± 14.3	153.0 ± 13.3	152.5 ± 15.5	76.5 ± 13.3	76.0 ± 12.6	73.8 ± 13.1	0.572	0.02	<0.001
1RM Squat mean velocity (m/s)	0.16 ± 0.04	0.17 ± 0.02	0.17 ± 0.03	0.18 ± 0.02	0.19 ± 0.02	0.19 ± 0.02	0.811	0.157	0.025
Repetitions to failure with 70% of 1RM in the squat (repetitions)	21.2 ± 5.7	17.2 ± 2.7	16.6 ± 2.3	20.1 ± 4.7	16.0 ± 4.2	15.0 ± 4.7	0.954	<0.001	0.43
CMJ (cm)	43.5 ± 5.9	39.5 ± 7.3	39.5 ± 5.8	31.5 ± 4.6	28.0 ± 5.7	27.8 ± 5.4	0.933	<0.001	<0.001
Session RPE	6.6 ± 1.1	6.0 ± 1.2	5.9 ± 0.9	6.0 ± 0.7	5.5 ± 1.0	5.9 ± 1.0	0.374	0.054	0.303

Data are reported as mean ± standard deviation; 1RM—one-repetition maximum; RPE—rating of perceived exertion.

**Table 2 nutrients-14-04840-t002:** Effect sizes (Hedges’ *g*) for pairwise comparisons for the variables in which there was a significant main effect of caffeine.

Variable	Caffeine vs. Placebo	Caffeine vs. Control	Placebo vs. Control
1RM bench press (kg)	0.08 (95% CI: 0.03, 0.13)	0.12 (95% CI: 0.08, 0.17)	0.04 (95% CI: 0.01, 0.08)
Repetitions to failure with 70% of 1RM in the bench press (repetitions)	1.41 (95% CI: 0.91, 2.03)	1.28 (95% CI: 0.80, 1.86)	−0.12 (95% CI: −0.46, 0.21)
1RM squat (kg)	0.03 (95% CI: −0.01, 0.07)	0.06 (95% CI: 0.02, 0.10)	0.03 (95% CI: −0.02, 0.08)
Repetitions to failure with 70% of 1RM for squat (repetitions)	0.89 (95% CI: 0.36, 1.47)	1.04 (95% CI: 0.59, 1.57)	0.21 (95% CI: −0.12, 0.55)
CMJ (cm)	0.43 (95% CI: 0.23, 0.66)	0.46 (95% CI: 0.24, 0.72)	0.02 (95% CI: −0.14, 0.17)

1RM—one repetition maximum; CMJ—countermovement jump; CI—confidence interval.

**Table 3 nutrients-14-04840-t003:** Blinding was assessed before and after the exercise tests.

	Recreational				Trained			
	Pre-Test		Post Test		Pre-Test		Post-Test	
	PLA	CAF	PLA	CAF	PLA	CAF	PLA	CAF
Guessed CAF	3	2	2	3	0	3	0	4
Guessed PLA	3	1	3	0	3	2	3	1
Did not know	4	7	5	7	7	5	7	5

CAF: caffeine; PLA: placebo.

## Data Availability

The data presented in this study are available on request from the corresponding author. The data are not publicly available due to privacy restrictions.

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
