# Peer review of "Are Caffeine’s Effects on Resistance Exercise and Jumping Performance Moderated by Training Status?"

_nutrients, 2022, doi:10.3390/nu14224840_

Round 1

Reviewer 1 Report

A well planned and executed study. The write up of the study was excellent. 

Author Response

Comments and Suggestions for Authors

A well planned and executed study. The write up of the study was excellent. 

Response: Thank you very much for your time and positive comments.

Reviewer 2 Report

In general, this is a well-written manuscript, with interesting results, and I enjoyed reading it. However, there are some important issues that need to be addressed and improved. Major and minor points as follows:

Major:

1.       Details of the participants in each group need to be given, not just all combined – need to show that there was no significant difference between groups at the start – in relation to age and BMI.

2.       The statistical results within groups needs to be shown. For example - the authors give the results of statistical analysis between caf, con and pla, but this seems to be for both groups (trained and untrained) combined. The same pattern of asterisks is used in both trained and untrained graphs, which may not be a correct representation. The graphs should represent statistical analysis done within each group – ie. separate analysis for trained and untrained. That would be more correct and it would also highlight any subtle differences between trained and untrained. The data in Table 1 is very clear – inclusion of the statistical analysis on that table would be helpful.

3.       Where statistics are used to compare trained vs recreationally active, it would be better to represent both on one graph, where they could be compared visually using one axis. Currently the two graphs use two different axes and the significant difference is not clearly represented.  

4.       More detail on the caffeine tablet: Did the investigators make these capsules themselves, with specific weight designed for each participant? Item C0750 is available as a powder from Sigma-Aldrich. If so, what other ingredients are in the capsules, and how are they produced?

 Minor:

5.       Clarity and consistency in language: ‘Condition’ is used in results to refer to caffeine/control/placebo but it would be better to use ‘caffeine’ or ‘caffeine intake’ in all instances to be consistent and avoid confusion. Also, ‘untrained’ vs ‘recreationally active’.

6.       Regarding the statement from line 343: ‘Specifically, while we did not find that training status modulated the effects of caffeine, it might be that highly trained athletes may not benefit from caffeine ingestion, as their performance is already at a high level and might be approaching their genetic ceiling’. The findings do not really support this theory, as caffeine caused an increase in the trained group also. Correct reporting of stats within groups as mentioned above may provide more clarity on this.

7.       Definitions of trained vs untrained participants need to be better clarified in the methods: ‘The recreationally active had less than 6 months resistance training and also had 3 or fewer hours of this activity per week’ – is that 3 or fewer hours of resistance training or sport-specific training? The trained may have also only had 3 hours of training – they trained 3 times/week in a sport. Also, the minimum muscular strength for the trained groups is given, but this detail is not given for the untrained – (squat with 1.5 times body weight and bench press equal to body weight) group? The difference is mentioned later in the discussion, but needs to be clearly stated in the methods.

Author Response

Comments and Suggestions for Authors

In general, this is a well-written manuscript, with interesting results, and I enjoyed reading it. However, there are some important issues that need to be addressed and improved. Major and minor points as follows:

Response: Thank you very much for your time and constructive feedback. Our responses to specific comments are provided below.

Major:

  1. Details of the participants in each group need to be given, not just all combined – need to show that there was no significant difference between groups at the start – in relation to age and BMI.

Response: Per suggestion, we provided age, height, body mass, and BMI for each group individually. There was no significant difference (p>0.05) for these participant characteristics.

  1. The statistical results within groups needs to be shown. For example - the authors give the results of statistical analysis between caf, con and pla, but this seems to be for both groups (trained and untrained) combined. The same pattern of asterisks is used in both trained and untrained graphs, which may not be a correct representation. The graphs should represent statistical analysis done within each group – ie. separate analysis for trained and untrained. That would be more correct and it would also highlight any subtle differences between trained and untrained. The data in Table 1 is very clear – inclusion of the statistical analysis on that table would be helpful.

Response: We thank the reviewer for the comment. We only performed within-group post hoc analyses if there was a significant interaction effect. Given that there were no significant interaction effects for any of the analyzed outcomes, we did not perform post hoc analyses for the individual group. Given that for the majority of outcomes that was a significant main effect, explored main effect we performed a pairwise post hoc (Tukey’s) to compare the effects between the three conditions (i.e., caffeine vs. placebo, caffeine vs. control, and placebo vs. control). If there was a main effect of training status, we simply reported the p-value since there were only two groups and no post hoc analysis was required. The figures were created to demonstrate the individual responses, however, we agree that this is confusing based on the statistics and we have removed the figures for clarity. Furthermore, we did descriptively report the means and standard deviations in table 1 and further explored hedges g (only if statistically appropriate (i.e., in a case of a significant main effect)).

  1. Where statistics are used to compare trained vs recreationally active, it would be better to represent both on one graph, where they could be compared visually using one axis. Currently the two graphs use two different axes and the significant difference is not clearly represented.  

Response: As previously mentioned, since the figures did not match the corresponding statistical analysis, we have selected to remove the figures for clarity.

  1. More detail on the caffeine tablet: Did the investigators make these capsules themselves, with specific weight designed for each participant? Item C0750 is available as a powder from Sigma-Aldrich. If so, what other ingredients are in the capsules, and how are they produced?

Response: We have added in information about the item C0750 which was pure caffeine anhydrous.

Line 137-142: Participants ingested one capsule containing 6 mg/kg body weight of pure caffeine anhydrous (Cat. No. C0750; Sigma-Aldrich; Steinheim; Germany) or placebo (cellulose) 60 minutes before starting the warm-up for each participant [27]. A high-precision electronic digital scale was used to weigh the caffeine and placebo pow-der, which was administered to capsules by an independent researcher.

 Minor:

  1. Clarity and consistency in language: ‘Condition’ is used in results to refer to caffeine/control/placebo but it would be better to use ‘caffeine’ or ‘caffeine intake’ in all instances to be consistent and avoid confusion. Also, ‘untrained’ vs ‘recreationally active’.

Response: We have revised accordingly. “Caffeine” and “recreationally active” were used instead of condition in the results.

  1. Regarding the statement from line 343: ‘Specifically, while we did not find that training status modulated the effects of caffeine, it might be that highly trained athletes may not benefit from caffeine ingestion, as their performance is already at a high level and might be approaching their genetic ceiling’. The findings do not really support this theory, as caffeine caused an increase in the trained group also. Correct reporting of stats within groups as mentioned above may provide more clarity on this.

Response:  As previously mentioned, we only found a main effect of condition and there was no significant interaction effect. Given the lack of an interaction effect, we did not perform separate analyses for both groups. Still, we agree that this sentence might have been confusing and have modified it for clarity. 

  1. Definitions of trained vs untrained participants need to be better clarified in the methods: ‘The recreationally active had less than 6 months resistance training and also had 3 or fewer hours of this activity per week’ – is that 3 or fewer hours of resistance training or sport-specific training? The trained may have also only had 3 hours of training – they trained 3 times/week in a sport. Also, the minimum muscular strength for the trained groups is given, but this detail is not given for the untrained – (squat with 1.5 times body weight and bench press equal to body weight) group? The difference is mentioned later in the discussion, but needs to be clearly stated in the methods.

Response: We thank the reviewer for the comment. The recreationally active participants performed up to three hours of this activity per week. On the other hand, the trained participants generally resistance-trained four or more hours per week. Regarding the criteria of muscular strength, we did not use any specific criteria for recreationally active participants. However, given their limited experience in resistance training, we expected low levels of muscular strength in the recreationally active participants. Our analysis has shown this, given that higher muscular strength levels were found in the trained cohort.

Round 2

Reviewer 2 Report

Thank you for addressing the issues raised in the first review. I appreciate that the authors have made changes to clarify the results, but I think it would be useful to indicate statistics on the table. I also think that it would be of interest to include the within-group ANOVA as well as the overall 2 way ANOVA, even if the results are not significant. However, I accept that the main questions are covered by the 2 way ANOVA.

In relation to the trained vs untrained – although not statistically significant, the trained group appear to be slightly older and physically bigger (which may or may not be as a consequence of training). While it is stated that there is not sig difference between the groups, the statistical test used should be specified. Also, despite the lack of statistical difference between them, even a small physical difference between the groups may be relevant to the results and may be worthwhile mentioning in the discussion.

Author Response

Thank you very much for your comments and suggestions. We have addressed them point by point below.

Thank you for addressing the issues raised in the first review. I appreciate that the authors have made changes to clarify the results, but I think it would be useful to indicate statistics on the table. I also think that it would be of interest to include the within-group ANOVA as well as the overall 2 way ANOVA, even if the results are not significant. However, I accept that the main questions are covered by the 2 way ANOVA.

Response: As suggested, we have inserted the p-values into the table. As for the within-group ANOVA, this test would only be performed if a significant interaction was found. Given that there was no significant interactions, we did not perform any further analyses. Forcing a within-subject ANOVA with no interaction is statistically incorrect and greatly increases the likelihood of a type I error.

In relation to the trained vs untrained – although not statistically significant, the trained group appear to be slightly older and physically bigger (which may or may not be as a consequence of training). While it is stated that there is not sig difference between the groups, the statistical test used should be specified. Also, despite the lack of statistical difference between them, even a small physical difference between the groups may be relevant to the results and may be worthwhile mentioning in the discussion.

Response: We have clarified in the methods that a t-test was used to compare participant characteristics between trained and recreational groups. In addition, based on previous research caffeine’s efficacy is not influenced by age, so this would no be a confounding factor. Furthermore, caffeine doses were provided relative to body weight, so differences in body mass would also not confound the results.